# On-Site Multisample Determination of Chlorogenic Acid in Green Coffee by Chemiluminiscent Imaging

**DOI:** 10.3390/mps6010020

**Published:** 2023-02-14

**Authors:** Sergi Mallorca-Cebria, Yolanda Moliner-Martinez, Carmen Molins-Legua, Pilar Campins-Falcó

**Affiliations:** MINTOTA Research Group, Departament de Química Analítica, Facultad de Química, Universitat de Valencia, C/Doctor Moliner 50, 46100 Burjassot, Spain

**Keywords:** antioxidants, chlorogenic acid, chemiluminiscence, CCD camera, green coffee, on-site

## Abstract

The potential of antioxidants in preventing several diseases has attracted great attention in recent years. Indeed, these products are part of a multi-billion industry. However, there is a lack of scientific information about safety, quality, doses, and changes over time. In the present work, a simple multisample methodology based on chemiluminiscent imaging to determine chlorogenic acid (CHLA) in green coffee samples has been proposed. The multi-chemiluminiscent response was obtained after a luminol-persulfate reaction at pH 10.8 in a multiplate followed by image capture with a charge-coupled device (CCD) camera as a readout system. The chemiluminiscent image was used as an analytical response by measuring the luminescent intensity at 0 °C with the CCD camera. Under the optimal conditions, the detection limit was 20 µM and precision was also adequate with RSD < 12%. The accuracy of the proposed system was evaluated by studying the matrix effect, using a standard addition method. Recoveries of chlorogenic acid ranged from 93–94%. The use of the CCD camera demonstrated advantages such as analysis by image inspection, portability, and easy-handling which is of particular relevance in the application for quality control in industries. Furthermore, multisample analysis was allowed by one single image saving time, energy, and cost. The proposed methodology is a promising sustainable analytical tool for quality control to ensure green coffee safety through dosage control and proper labelling preventing potential frauds.

## 1. Introduction

The increasing consumption of antioxidants raises public health concerns about their efficiency, safety, and real benefits for human health. Antioxidants are widely available and commonly used, especially in the women’s health market [1]. However, there is a lack of information about the composition, doses, potential degradation, and function of antioxidants [2]. Mainly, the benefits of antioxidant supplements are associated with their interaction with free radicals and hence protect health against oxidative stress [3]. The interaction mechanism depends on the chemical nature and reactivity. These compounds are classified as direct and indirect antioxidants and they mediate oxidative stress through different mechanisms. Nevertheless, there are controversial results and conclusions about the real level and function of these compounds [4].

What is certain, is that the antioxidant nutraceutical market is now a multi-billion dollar industry experiencing a continuous increase, and this has led to regulations that ensure safety, proper labelling, and more information about the ingredients, which means there is a real need to ensure the concentration of the antioxidant compounds, and that they are within safety limits, in particular in quality control studies [5].

Focusing on direct antioxidants, chlorogenic acid (CHLA) is an abundant water-soluble antioxidant in the human diet. Indeed, it is available from a wide variety food and beverage sources. In particular, CHLA is linked to green coffee and green coffee extract-based products’ health benefits. That fact, together with the relevance of the global coffee market, has led to a growing interest in the development of analytical methods for the quantification of this antioxidant [6].

In this context, the development of analytical strategies that facilitates the determination of CHLA, in commercial products is mandatory to help the regulation, labelling, and warming of potential side effects [7]. Basically, the main methods to determine this compound are based on separation techniques such as HPLC and UPLC, coupled to different detectors such as DAD, luminescent, and MS [8,9,10,11,12]. Their performance has been demonstrated, yet they required specialized laboratory instrumentation, trained personnel, and a large instrumental investment. Thus, more recently simple, cost-effective, rapid, easy-handling, and portable analytical procedures have been proposed.

These new strategies are principally based on optical readouts with the added value of providing visual responses that allow semiquantitative determination by visual or image inspection and on-site determination [13]. Colorimetric-based sensors using paper-based devices and a metal nanooxide array have been recently published [14,15].

Among the different on-site strategies, chemiluminiscence (CL) is attracting attention since it allows for the achievement of sensitivity in short analysis times [16]. Going a step forward, chemiluminiscent imaging is a powerful alternative since it combines the typical CL properties with visual inspection and portability [12,17]. In this case, the use of charge-couple device (CCD) cameras has been demonstrated to be a feasible detector for this aim [18]. Fundamentally, the photons produced in the hemiluminescent reaction are captured by the CCD camera and subsequently, an image is displayed based on the light generated, the intensity measurement can be correlated with the target analyte. The key parameters in the application of these systems are mainly based on the control of experimental variables since the CL efficiency depends on the kinetic emission CCD camera response time. From the analytical point of view, there is scarce knowledge of the potential application of CCD cameras as analytical response read-out systems. This strategy has been proposed for environmental analysis [18], however, other application areas and in particular for quality control of human consumption products, are still unexplored.

The objective of the present work was to demonstrate the utility of chemiluminiscent imaging employing the CCD camera to estimate CHLA in natural products, in particular green coffee samples. The novelty relies on the combination of an on-site portable device with high-performance analytical parameters for quality control of commercial products. Optimization of the reaction parameters was aided by the use of a portable luminometer. For this aim, the instrumental parameters were optimized and the analytical properties were established. Finally, the practical application and validation of the proposed methodology were evaluated by analysing commercial samples.

## 2. Experimental Section

### 2.1. Reagents

Sodium hypochlorite 10% was obtained from Panreac (Barcelona, Spain). Sodium persulfate, Trolox, CHLA, and luminol were purchased from Sigma-Aldrich (Taufkirchen, Germany). Sodium carbonate and sodium hydroxide were from VWR Chemicals (Radnor, PA, USA).

Standard solutions of sodium persulfate (50 mM) in carbonate buffer 0.3 M (pH 10.8) were prepared weekly. Trolox (1.1 mM), luminol (20 mM), and CHLA (2.0 mM) were prepared in carbonate buffer (0.3 M, pH 10.8). Working standard solutions of Trolox and CHLA were prepared by adequate dilution in carbonate buffer.

### 2.2. Equipment

The CL response was measured by using a 338L Mono CCD Camera themoelectrically cooled at 5 °C (Atik Cameras, Norwich, UK. In addition, an LB 9509 portable tube luminometer (Berthold Technologies, Bad Wildbad, Baden-Württemberg, Germany) was also used to determine the optima chemiluminiscent reaction conditions.

### 2.3. Procedure

#### 2.3.1. Optimization and Measurement of the CL Response

In the preliminary studies, the optimization of the reaction parameters, such as oxidant, concentration, pH, and analysis time was carried out using the luminometer and CHLA analytical grade standards (see Section 2.1). Oxidants such as sodium persulfate and sodium hypochlorite were evaluated. For this aim, 100 µL of luminol was mixed with 100 µL of persulfate or hypochlorite of luminol and the response was registered every 5 s, for 1 min. The study of persulfate and luminol concentrations were performed in the range of 0.7–10 mM, and 1.4–4.8 mM, respectively, by measuring the signal at 60 s. Finally, the influence of pH was studied in the interval 10.0–11.5, using persulfate (10 mM) and luminol (4.8 mm).

Once the optimal reaction conditions were established, the CCD camera instrumental parameters were studied to monitor the CL imaging response.

#### 2.3.2. CCD Measurements

The CCD camera was studied as a readout system. In this case, contrast, temperature, and exposition time were previously optimized. Under the different conditions, the images were processed by the software Artemis Capture, and the luminescent intensity (I) for each spot was registered. Standards and samples were deposited on a multiplate and placed inside the CCD camera dark chamber for its measurement. The device used to perform the CCD measurements is shown in Figure 1. It should be noted that exposition distance was optimized in previous work [18].

The measurement was performed in a multiplate by mixing luminol (100 µL) and persulfate (100 µL). After 5 min, 100 µL of the blank, working standards, or samples, prepared in a carbonate buffer (0.3 M, pH 10.8), were added, and the image was registered. The working concentration interval was up to 500 µM for CHLA. Trolox was also measured as another example of the antioxidant analyte.

### 2.4. Analysis of Commercial Green Coffee Samples

The application of the proposed method was validated by the analysis of three commercial green coffee samples acquired in a local store. Sample preparation was carried out as follows: a capsule of each sample was dissolved in ultrapure water and sonicated during 5 min, to extract CHLA. Subsequently, the extract was diluted in carbonate buffer (0.3 M, pH 10.8). Samples were measured following the proposed procedure. The analysis was performed in triplicate. In addition, the accuracy of the proposed method was evaluated by using the standard addition method (SAM). For this aim, the calibration curve was performed following the procedure described in Section 2.3 and spiking the sample with known concentrations of CHLA standards.

## 3. Procedure

### 3.1. Study of Chemiluminiscent Imaging as an Analytical Response

In the previous experiments, the CL response was optimised considering the hypothesis that the reaction between luminol and an oxidant in basic media, induces the generation of a luminescent signal that can be monitored, and the presence of antioxidants such as chlorogenic acid may inhibit the CL signal as a function of the concentration. Hence, the first step was the optimization of the CL reaction, taking into account that the CCD camera will be the readout system, which means that detection time is a key parameter. Figure 2 shows the variation of the CL response as a function of the experimental parameters (measurements done with the portable luminometer).

As can be seen in Figure 2A, the use of hypochlorite as an oxidant resulted in a fast reaction kinetic with a signal decay in 10 s. Meanwhile, persulfate provided a constant CL signal up to 60 s. Therefore, persulfate was selected as an oxidant, taking into account the exposition time that subsequently will be needed for the measurement in the CCD camera. Regarding luminol and persulfate concentrations (see Figure 2B,C), 4.8 mM, and 10 mM were the optimum concentration, respectively, since they maximized the CL signal, and hence the sensitivity of the target analytes by inhibition of this CL response. It is well known the dependence of CL reactions with pH, thus, this parameter was also studied. NaOH and carbonate buffer were studied. The results indicated that carbonate buffer 0.3 M and pH 10.8, provided satisfactory intensity, taking into account the concretion level of the samples of interest.

Under the above-mentioned conditions, the influence of the addition of antioxidants was evaluated. Previous reports indicated that antioxidants give rise to a decrease in the CL response and the mechanism involved is not yet clear, since some authors pointed to CL quenching in addition to the inhibition of luminol oxidation by the antioxidants that has also been reported [17]. In this scenario, different strategies were evaluated. In the first case, persulfate and CHLA were mixed, and after 10 min, luminol was added. In the second experiment, luminol and persulfate were mixed, and after 5 min, CHLA was added. Figure 1D shows the results obtained. As can be seen, in the second condition, a decrease of the CL signal was observed with the addition of CHLA. Hence, that was the strategy selected for further experiments.

By another hand, the variation of the CL decrease as a function of CHLA concentration was also studied in order to establish the potential use as a quantitative methodology. Firstly, the reaction time between persulfate and luminol, before the addition of the antioxidant was studied. For this aim, additional times of 180 and 300 s were studied. Figure 3A shows the variation of the response as a function of time, and at two antioxidant addition times. The addition of the target analytes at 300 s was advantageous since the stability of the response was higher than at 180 s, more likely due to a slower kinetic of the antioxidants since, persulfate in the reaction media is lower at 300 s than at 180 s. This stability was a key parameter, since, the CCD camera required a stability interval to properly register the image. The inclusion criteria, in this case, was that the signal was stable for up to 30 s. Figure 3B shows the variation of the kinetics at different concentration levels. By measuring the signal at 30 s after the addition of chlorogenic acid, the CL decrease could be linearly correlated with the concentration level (sensitivity = −1400 ± 50 µM^−1^, R^2^ = 0.9930), hence the quantitative potential was demonstrated under these conditions with a detection limit, LOD = 2 µM.

Remarkably, it should be considered that in presence of other antioxidant compounds, the decrease in the luminescent signal would be a result of the total antioxidant content extracted under the extraction conditions described in Section 2.4. In order to demonstrate this, the RLU signal for trolox was also measured. The results indicated the same behaviour as CHLA, however, the decrease of the luminescent signal was slightly higher, achieving a detection limit of 0.7 µM. Hence, in this case, the luminescent signal would provide a measurement of the total content of CL active compounds under the extraction conditions.

Once the reaction parameters were optimized, the CCD camera instrumental parameters, the contrast parameters, exposition time, and temperature were studied. The CL response was established by capturing the image and processing that image with Artemis Capture software. Figure 4 shows the schematic representation of the set-up and the images obtained for different blanks measured at different times under the contrast parameters white 2000, black 450, and log 1.5 for CHLA. It should be noted that the image shown in Figure 4 was a real image, where the responses were specific dots for each blank. In this case, the analytical response was the luminescent intensity (I) of each sample, and hence the balance between black and white intensity and grey contrast was fundamental to obtain adequate image quality. As can be seen, the profile of the CL image signal was similar to that measured with the luminometer. Hence, it was expected, that the addition of CHLA to the plate would result in a decrease in the CL that can be monitored by the CCD camera.

To demonstrate the response as a function of CHLA concentration, the target analyte was added and the experiment was performed at two temperatures. The results indicated that the temperature of the CCD camera was a key parameter. Mainly at high a temperature, the background noise or dark current gave rise to a low-resolution image. Meanwhile, working at 0 °C, the resolution was satisfactory, and the difference in the blank signal compared with the CHLA standard could be monitored by the image. Figure 5A shows the CL images carried out at 0 and 23 °C for CHLA compared with blanks at the same temperature.

The variation of the luminescent intensity with the exposition time can be seen in Figure 3, the readout measurement is stable up to 30 s due to the chemistry of the reaction. Hence, exposition times from 5 to 20 s were studied. The results indicated, that at times lower than 10 s, the image resolution was poor. At exposition times higher than 10 s, the resolution was satisfactory, however, there was no improvement of the longer times (20 s) used. Therefore, 10 s was selected for further experiments.

Finally, and under the above-mentioned instrumental parameters, the variation of the CCD camera response was studied as a function of the concentration of CHLA. Figure 5B shows the image obtained. These results clearly demonstrated that the decrease of the CL signal as a function of the concentration of CHLA was linearly correlated (Figure 5C). It should be noted that the image parameter black and white, which enhanced the intensity of black and white, and the grey contrast up to 1.5 (log parameter) gave rise to an adequate image quality that allowed the determination of the target analyte up to 400 µM.

### 3.2. Analytical Parameters of the Procedure

Table 1 summarizes the analytical parameters such as sensitivity as the slope of the calibration curve, linearity, and detection limit. Moreover, intra-and interday precision was also evaluated as the relative standard deviation (%RSD)

The results indicated that when using a chemiluminiscent image as an analytical response, the working interval was up to 400 µM, and the limit of detection was 20 µM. This LOD was satisfactory for the application in dietary products, however, a more sensitive determination will require the use of other analytical responses, such as the use of a portable luminometer. Intraday and interday precision was also evaluated. In both cases, satisfactory RSD values were achieved for practical applications.

As mentioned before, previous methods based on HPLC have been proposed, and the comparison reveals that, in general, chromatographic methods provided LOD slightly lower than the LOD obtained with the CCD method, recoveries were also satisfactory, and, obviously, selectivity was better when using separation methods [19]. Capillary electrophoresis with CL detection has resulted in an adequate method for the analysis of natural products, however, the LOD was higher than that obtained with the proposed method [20]. NIR has also been proposed for CHLA determination, with successful results. However, multivariate calibration was necessary [21]. Regarding the in situ-based analytical systems, sensors and biosensors have also been studied in depth. A review focused on this topic has been recently published by Munteanu and Apetrei [6]. Recent advances in sensitive materials for electrochemical sensors and biosensors have led to methodologies, with low µM detection limits, which is an advantage if the content of this type of antioxidant is at a trace level [22]. However, precision depends on the accuracy and expertise in the preparation of the sensitive material. Therefore, the proposed methodology shows some advantages over these previously proposed methods; since the determination of CHLA can be performed by image inspection, a semiqualitative analysis for sample screening is possible. In addition, quantitative analysis can be also carried out for samples with contents higher than 20 µM. By another hand, cost of the CCD is lower than chromatographic electrophoretic methods.

### 3.3. Procedure Validation: Application to the Analysis of Green Coffee Samples

In order to validate the proposed procedure for its practical application, the analysis of three different green coffee samples, being CHLA the active compound. In addition, and in order to evaluate the matrix effect and the potential interference of other ingredients present in the sample composition, the standard addition method (SAM) and a recovery study were carried out and the slopes of the calibration curves were compared with external calibration. Figure 6 shows the images of both calibration strategies. As was expected, the luminosity intensity of the blank in the external calibration was higher than in the first sample of the standard addition method due to the presence of CHLA, and in both cases, there was a decrease in the CL intensity over the CHLA concentration.

The slopes of the SAM calibration were −0.49 ± 0.3, −0.37 ± 0.02, −0.37 ± 0.01 µM^−1^ for samples 1, 2, and 3, respectively. The statistical analysis of the SAM slope and the slope of the external calibration (see Table 1) demonstrated that the sample matrix may influence the luminescent intensity, and hence, it may be necessary for SAM as a calibration method. In the samples analysed in this work, only sample 1 was not affected by the matrix effect, since there were no statistical differences in the slope of the calibration graph. However, that influence depended on the commercial product, the matrix effect was observed in samples 2 and 3. The calculated CHLA contents were 145 ± 10, 220 ± 12, and 204 mg CHLA/ capsule for sample 1, sample 2, and sample 3, respectively. These results were then compared with the labels of the commercial products. Sample 1 and sample 2 were labelled with a CHLA content of 157 mg/capsule and 234 mg/capsule, respectively. The statistical comparison with the quantitative analysis carried out with the proposed methodology revealed no statistical differences, and hence, accuracy was demonstrated. In the case of sample 3, the content of CHLA was not labelled. However, the analysis of the sample determined 207 mg/capsule, expressed as CHLA. Recovery values were in the range between 93 ± 8 and 94 ± 9 %. These results indicated that the proposed methodology can be a relevant methodology to evaluate the content of CHLA in the analysed samples, since commercially, some of these products are not properly labelled, and that matrix effect has to be evaluated as a function of the analysed samples. It has been demonstrated that standard addition methods provided satisfactory results in the case of the matrix effect.

## 4. Conclusions

In this work, an on-site methodology based on chemiluminscent imaging has been proposed to estimate the concentration of CHLA in green coffee samples. The fundamentals of the proposed strategy relied on the use of the CL to decrease the presence of antioxidants, in particular CHLA, produced in the reaction luminol-persulfate under basic conditions. Under the optima chemical conditions, the results indicated that the addition of CHLA induced a stable analytical response that can be used as an analytical signal for the CCD camera as the detection system. The image taken by the CCD camera was processed. At a T = 0 °C and an exposition time of 10 s, the decrease in the luminiscent intensity was linearly correlated with CHLA concentration, with a LOD of 20 µM and RSD values up to 12%. The satisfactory analytical performance was validated by applying the proposed strategy for the analysis of commercial green coffee samples. The results demonstrated that the matrix effect should be considered when analysing real samples. SAM was adequate for the calibration since the sample matrix may influence the luminosity intensity, however, that effect depended on the sample. The analysis of samples was validated by comparing the results with the labelled products, indicating that satisfactory accuracy was obtained. Hence, it can be concluded that the proposed strategy can be a potential on-site tool to estimate the concentration of antioxidants in dietary products. The analysis of the chemiluminiscent image in the set-up allowed the multisample detection resulting in a cost-and energy-efficient protocol with reagent consumption at µM level. That demonstrated that the proposed method is a promising sustainable analytical tool for quality control to ensure supplement safety and proper labelling, preventing potential fraud.

## Figures and Tables

**Figure 1 mps-06-00020-f001:**
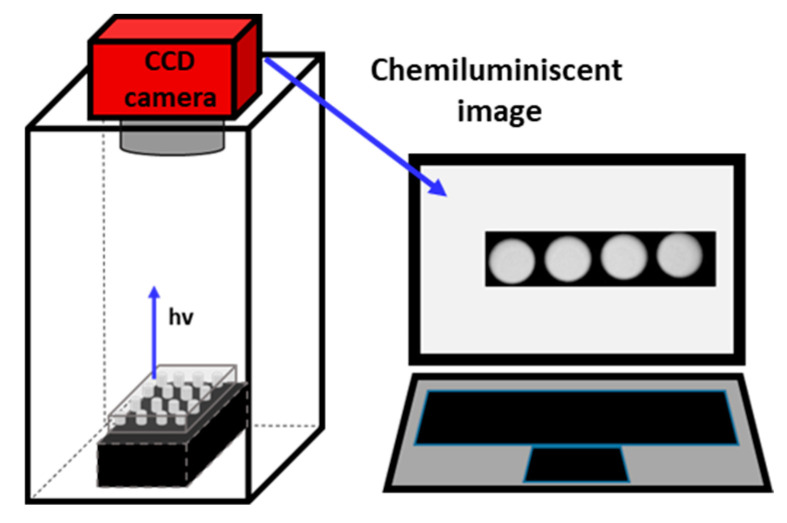
Schematic diagram of the CCD camera dark chamber used for the determination of CHLA.

**Figure 2 mps-06-00020-f002:**
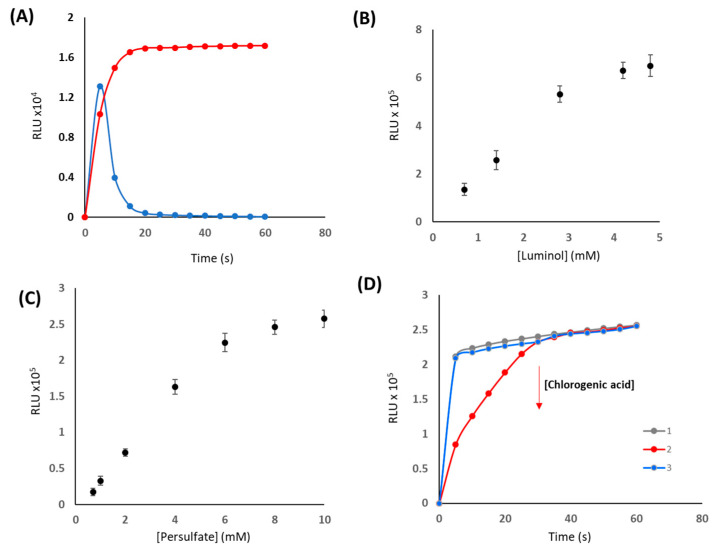
(**A**) Variation of the luminescent signal (RLU) produced by luminol using 0.25 mM hypochlorite (blue) and 0.7 mM persulfate (red); Luminol = 1.4 mM, carbonate buffer 0.3 M, pH 10.8. (**B**) Variation of the RLU as a function of luminol concentration, (**C**) Variation of the RLU as a function of persulfate concentration, (**D**) Variation of the RLU signal in presence of CHLA (50 µM, pH 10.8) for blank (1, grey), luminol-persulfate, CHLA addition at t = 5 min (2, red) and persulfate-CHLA luminol addition at t = 10 min (3, blue). Conditions: 10 mM persulfate, 1.4 mM luminol.

**Figure 3 mps-06-00020-f003:**
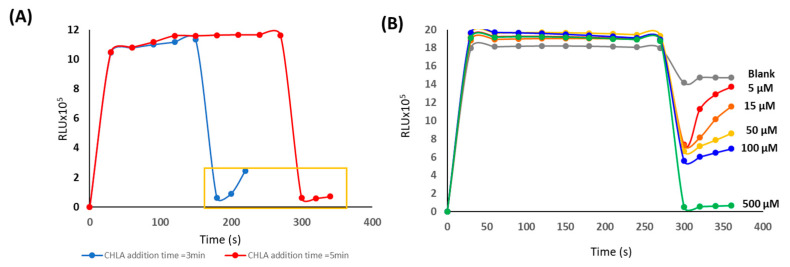
(**A**) Variation of the RLU in presence of CHLA (500 µM) after 180 s (blue) and 300 s (red). (**B**) Variation of the RLU over CHLA concentration under the optima reaction conditions.

**Figure 4 mps-06-00020-f004:**
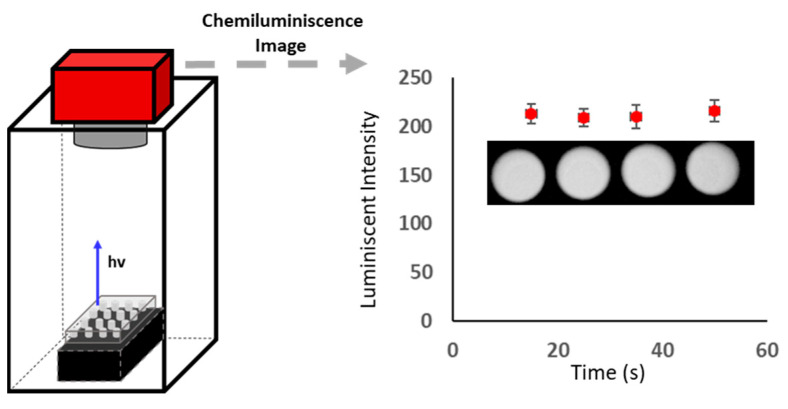
Schematic set-up of the CCD camera to measure CL image, the real image of the variation of the bank signal as a function of time, and the representation of the luminescent intensity for each image.

**Figure 5 mps-06-00020-f005:**
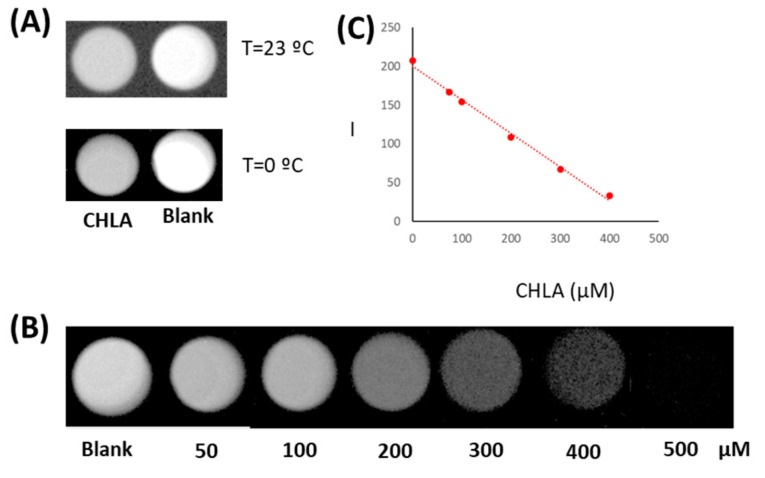
(**A**) Effect of the CCD camera temperature on the chemiluminiscent image (CHLA 100 µM), (**B**) Variation of the chemiluminiscent intensity over CHLA concentration, and (**C**) Correlation of the CL intensity as a function of CHLA concentration.

**Figure 6 mps-06-00020-f006:**
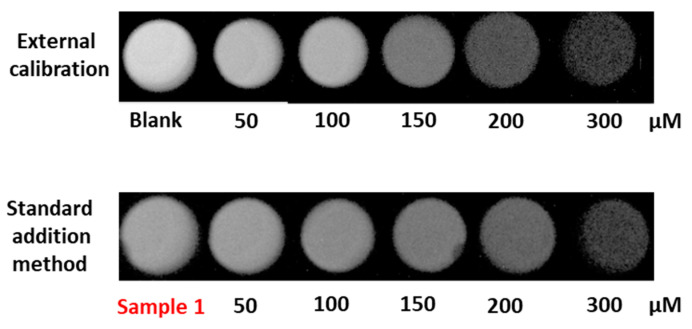
Comparison of the CL image for external calibration and standard addition method in sample 1.

**Table 1 mps-06-00020-t001:** Analytical parameters and precision (%RSD) values for the determination of CHLA by the CCD camera.

	Analytical Parameters
Working interval (µM)	50–400
a ± s_a_	200 ± 4
b ± s_b_ (µM^−1^)	0.470 ± 0.017
R^2^	0.999
LOD (µM)	20
RSD _intraday_ (%)	2.0
RSD _interday_ (%)	12.0

a: intercept; s_a_: standard deviation of a; b: slope, s_b_: standard deviation of b; LOD: detection limit, RSD: relative standard deviation.

## Data Availability

Data is contained within the article.

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
