# Peer review of "On-Site Multisample Determination of Chlorogenic Acid in Green Coffee by Chemiluminiscent Imaging"

_mps, 2023, doi:10.3390/mps6010020_

Round 1
Reviewer 1 Report
The manuscript by Mallorca-Cebria entitled “On-site multi-sample determination…” argues for a new technology or methodology as an analytical tool for quality control to ensure supplement safety and proper labelling preventing potential fraudulent material. There are several points requiring clarification.
How is safety being assessed? The proposed method states that concentrations will be estimated by the chemiluminescent method.
The abstract is vague and needs considerably more detail aside from the general description of the CD camera. Overall, the manuscript is vague and the point is largely unclear as to how this tool will be superior to others.
The title may be misleading. There are some 70,000 dietary supplements on the market and the term antioxidant is a fairly generic. Is this method being proposed for direct and indirect antioxidants? lipid-soluble and water-soluble antioxidant? High-molecular weight versus low molecular weight? enzymatic versus non-enzymatic, etc.?
Please provide information on how the chemiluminescence system itself would work regarding chemistry with myriad antioxidants. In figure 1, will multiple samples be in the chamber with a camera seemingly at a considerable distance collecting a signal? Will there not be interference from hv emission by all samples or will samples show up as specific dots as indicated in the image of the laptop?
What is CHLA and its relevance? It is not defined.
Discussion in section 2.4 states that 3 seemingly arbitrary products were obtained from a local store. Why not use authentic, purified standards in case your store selections are adulterated or impure or degraded? A capsule was dissolved in water but what if one or more ingredients are lipid-soluble? How is this being addressed?
The y-axis labels could be simplified in figure 2.
Author Response
Please, see the attach file with the Response to Reviewer`comments

Reviewer 2 Report
The reviewer appreciate the interest of the authors in the development of this manuscript. It is an interesting topic.
The main aim of the authors was to apply a new method for assessing the content of bioactive - antioxidant components in dietary supplement research. In my opinion, the method has potential; however, the manuscript itself still needs many improvements.
1. Line 28-30 - please add relevant literature reference
2. Line 60-61 in which environmental analysis?-please provide literature references, examples of such analyses using this method.
3. Were reagents other than chlorogenic acid used in optimising the method?
4. Table 1 Please explain the letter markers used
5. How does the newly applied method compare to already commonly used methods (please compare)?
6. No explanation of the abbreviation "CHLA"
7. In the title, the authors mention the application of the method to the determination of antioxidants in dietary supplements, however, I did not find results for this group of products in the paper. In addition, information appears about three samples of green coffee analysed. I understand that this is a natural raw material and not a dietary supplement? Furthermore, the preparation step of the samples for determination is missing in the paper.
The reviewer appreciate the interest of the authors in the development of this manuscript. It is an interesting topic. However, the manuscript should be completed with the indicated information. I suggest MAJOR REVISIONS.
Author Response
Please, see the attach file with the Responses to Reviewer`s Comments

Round 2
Reviewer 1 Report
The authors have revised the manuscript to effectively clarify several potential points of ambiguity. The graphics are clear, logical, and well-constructed. The modifications have markedly strengthened the manuscript. This reviewer has no further queries.
Reviewer 2 Report
After the applied revisions to the manuscript, I recommend the paper for publication in the journal.